# Scheimpflug Camera-Based Technique for Multi-Point Displacement Monitoring of Bridges

**DOI:** 10.3390/s22114093

**Published:** 2022-05-27

**Authors:** Lei Xing, Wujiao Dai, Yunsheng Zhang

**Affiliations:** School of Geosciences and Info-Physics, Central South University, Changsha 410083, China; 185001005@csu.edu.cn (L.X.); zhangys@csu.edu.cn (Y.Z.)

**Keywords:** Scheimpflug camera, computer vision, multi-point displacement monitoring, camera motion compensation, low cost

## Abstract

Owing to the limited field of view (FOV) and depth of field (DOF) of a conventional camera, it is quite difficult to employ a single conventional camera to simultaneously measure high-precision displacements at many points on a bridge of dozens or hundreds of meters. Researchers have attempted to obtain a large FOV and wide DOF by a multi-camera system; however, with the growth of the camera number, the cost, complexity and instability of multi-camera systems will increase exponentially. This study proposes a multi-point displacement measurement method for bridges based on a low-cost Scheimpflug camera. The Scheimpflug camera, which meets the Scheimpflug condition, can enlarge the depth of field of the camera without reducing the lens aperture and magnification; thus, when the measurement points are aligned in the depth direction, all points can be clearly observed in a single field of view with a high-power zoom lens. To reduce the impact of camera motions, a motion compensation method applied to the Scheimpflug camera is proposed according to the characteristic that the image plane is not perpendicular to the lens axis in the Scheimpflug camera. Several tests were conducted for performance verification under diverse settings. The results showed that the motion errors in *x* and *y* directions were reduced by at least 62% and 92%, respectively, using the proposed method, and the measurements of the camera were highly consistent with LiDAR-based measurements.

## 1. Introduction

Structural health monitoring runs through the entire life cycle of civil engineering structures, and displacement measurement is an important technique in structural health monitoring. Currently, many types of sensors can be used to measure structural displacements, such as linear variable differential transformers (LVDTs) [1], laser Doppler vibrometers (LDVs) [2], global navigation satellite systems (GNSS) [3,4,5], total stations and image assisted total stations (IATS) [6,7,8]. However, the application of these sensors has certain limitations. For example, both the LVDT and LDV are limited by the measurement distance, making them impractical for large-scale field measurements. The GNSS is limited by insufficient measurement accuracy for high dynamic responses of the structure; its real time accuracy only reaches the centimeter level [5]. The total station is a high-precision non-contact sensor and is widely recognized, but it cannot fulfill the multi-point measurement requirement. To overcome this shortcoming, the latest development of total stations called IATS integrates a robotic total station with image sensors, which contains the advantages of high precision and multi-point measurement. However, the high costs of IATS restrict its extensive application [7].

Vision-based sensors provide a cost-effective, simple alternative for non-contact displacement measurement, and have been applied to various fields of structural displacement measurements [9,10], for example, wind tunnel tests of high-rise buildings [11], vibrational displacement measurements [12], defect detection [13] of bridges and slope deformation monitoring [14]. To determine displacements on distant bridges using vision-based sensors, scholars have proposed many approaches and made some achievements. Examples include dynamic displacement monitoring of bridges and high-rise buildings based on the grey centroid method [15], high frame rate (HFR) monitoring with artificial targets [10] and deflection measurements of bridges using a novel laser and video-based displacement transducer [16]. Recent studies have focused on improving the practicability of visual techniques [17]. For example, highly robust target localization methods have been designed to cope with complex illumination conditions [18]; additional sensors, such as the laser collimator [19] and total station [20], have been employed to compensate for camera motions [21]; some valuable investigations have been conducted to model or correct the thermal effect on camera image sensors [22,23].

Although scholars have proposed many practical methods for various challenges in the field, there are still many difficulties in applying visual techniques to bridge displacement monitoring. One of the main problems is that it is quite difficult to simultaneously measure displacements at many points on a bridge of dozens or hundreds of meters. Several multi-camera approaches [24,25,26,27] have been reported. Generally, a multi-camera system can be regarded as a combination of multiple single-camera systems, where each camera measures some points at different regions of the structure surface. Although a multi-camera system produces a sufficiently large effective field of view, the installation of multiple cameras is cumbersome and time-consuming. Moreover, with an increase in the number of cameras, the cost and uncertainty of the system increase. Therefore, it is still of practical significance to study single-camera methods. Aliansyah et al. [28] proposed installing a single camera at the front of the bridge to enable the measurement points fixed along the road direction of the bridge to be observed in a single FOV with a high-power zoom lens. However, for conventional cameras, the small DOF at high magnifications becomes problematic when focusing all points that lie in or close to a plane that is not parallel to the image plane.

In this work, an alternative approach of the Scheimpflug camera-based method for the multi-point displacement monitoring of bridges was proposed. The Scheimpflug camera, adopting the Scheimpflug condition by tilting the lens with respect to the image plane, enables the extension of the DOF of the camera without reducing the lens aperture and magnification; thus, it makes it possible to place only one camera at the front of the bridge to capture clear images of all measurement points distributed along the road direction of the bridge. Scheimpflug cameras have been applied to 3D digital image correlation [29], line structured light [30] and other fields [31]. However, to the best of our knowledge, the Scheimpflug camera-based method is rarely used in the literature on the multi-point displacement monitoring of bridges.

The remainder of this paper is organized as follows: Section 2 describes the limitations of multi-point displacement measurement using a conventional camera in detail and introduces the configuration and algorithms of the multi-point displacement monitoring of bridges using a single Scheimpflug camera. In Section 3, three tests are included. The first test evaluates the performance of camera motion compensation in the method with the help of a slide table. The second test evaluates the robustness of the method for long-distance measurements in outdoor environments. The third test is conducted on a truss structure bridge model and demonstrates the applicability of the proposed method and Scheimpflug camera-based system. Section 4 discusses some practical issues when applying the Scheimpflug camera to actual bridge monitoring, and Section 5 concludes this paper.

## 2. Materials and Methods

### 2.1. Limitations of Multi-Point Displacement Measurement Using Conventional Camera

Vision-based measurement systems have been widely used for defect detection and displacement measurements of bridges [32]. However, the following limitations remain to be solved for the multi-point displacement monitoring of bridges.

#### 2.1.1. The Contradiction between a Wide FOV and High-Resolution

The simple method of acquiring multi-point displacement data on a bridge is to install the camera at the side of the bridge to capture side-view images of the bridge, and the camera magnification should be reduced to observe all points in a single camera view. However, because of the limited number of pixels integrated on the image sensor, the reduction of camera magnification indicates a decrease in image resolution, thus affecting the accuracy of the displacement measurement, as illustrated in Figure 1a. 

#### 2.1.2. Narrow DOF at High Magnification

Generally, there are large inaccessible areas on the side of the bridge, whereas the front of the bridge is open and accessible along the road direction. Therefore, installing the camera at the front of the bridge and capturing measurement points arranged along the road direction in a single front-view without reducing the magnification has become the main mode of bridge displacement measurement. However, because of the narrow DOF of conventional cameras at high magnifications, capturing all measurement points clearly in a single front-view is quite difficult, as illustrated in Figure 1b. Aliansyah et al. [28] considered that lens blur does not significantly reduce the localization accuracy of the target; however, this assumption is not always practically applicable. A small lens aperture helps to extend the DOF, but it produces dark images owing to insufficient incident light. Therefore, a better approach for extending the DOF of the camera without reducing the magnification and lens aperture is necessary.

### 2.2. Multi-Point Displacement Measurement of Bridges Using Scheimpflug Camera

#### 2.2.1. Scheimpflug Camera-Based Measurement System and Displacement Calculation Algorithm

The Scheimpflug principle states that the focus plane (the plane on which the camera is focused), thin lens plane and image plane intersect in a single line, which is called the Scheimpflug line (Figure 2). In this case, the DOF of the camera is extended. Based on this principle, a robust, high-precision and low-cost displacement measurement system was designed in this study, which can clearly observe all measurement points distributed along the depth direction in a single-camera view with high magnification. The system contains a Scheimpflug camera, a tripod, laptop PC for camera control and several targets; their placement in bridge monitoring is illustrated in Figure 3. The Scheimpflug camera was installed at a stable area in front of the bridge, and each target was installed outside the bridge for its pattern to face the longitudinal direction of the bridge. The pattern of the target had two cross-shaped corners; thus, the scale conversion factors (mm/pixel) could be calculated easily. The distribution of the Scheimpflug camera and all targets were approximately in a line. The target installed on the stable platform (usually the pier) of the bridge was used as a reference for compensating camera motions. To facilitate the description of the algorithm in Section 2.2.2, the reference targets on two adjacent piers and the measuring targets between them were defined as a measuring unit, as shown in Figure 3.

The Scheimpflug camera used in this study includes three components: an 8-bit CMOS sensor employed to record the target images, which has a spatial resolution of 4096 × 2160 pixels; a telephoto lens (focal length 135 mm); and a custom-made Scheimpflug adapter. The adapter was machined by a computer numerical control (CNC) system, which can tilt the sensor around the vertical axis, with a range of approximately ±10°. The expense of the Scheimpflug adapter is only $100. The horizontal (H), vertical (V) and depth (D) directions of the Scheimpflug camera are defined as shown in Figure 4.

The displacement calculation algorithm mainly includes three steps. At first, the image coordinates of the targets are detected. To improve the accuracy and robustness of the localization method of cross-shaped targets, the sub-pixel method proposed by Duda and Frese [33] is utilized in this paper. Then, the sub-pixel displacement in the image plane can be obtained by calculating the difference between the centers of the targets in the continuous images sequence. Finally, the scale conversion factors in the corresponding direction need to be solved to convert image displacement into physical displacement, which can be obtained by comparing the physical dimension of the target with the pixel dimension in the image plane. It is assumed that the camera optical axis is almost perpendicular to the target plane. Therefore, the horizontal scale conversion factor *s_x_* and the vertical conversion factor *s_y_* can be solved as Equation (1),
(1)$$s_{x}=s_{y}=\frac{D_{physical}}{d_{image}}.$$

Physical displacement (*M_x_*, *M_y_*) can be obtained by multiplying the corresponding scale conversion factors:(2)$$\left\{\begin{array}{c} M_{x}=s_{x}\times dI_{x}\\ M_{y}=s_{y}\times dI_{y} \end{array}\right..$$
where *dI_x_* and *dI_y_* are the horizontal and vertical displacements in the image plane.

#### 2.2.2. Motion Compensation of Scheimpflug Camera

When a camera is installed for monitoring a full-scale structure, unexpected camera motion is unavoidable. Even if the camera is firmly fixed at a stationary point, its self-weight induces an inevitable and gradual movement of the entire system. In addition, cameras may be shaken by strong winds or ground vibrations in the field. Thus, the compensation of camera motions is necessary to ensure the accuracy of the displacement measurement. At present, utilizing fixed reference targets [34,35] to compensate camera motion is the most common and practical method, but existing methods have not considered the case that the image plane is not perpendicular to the lens axis in the Scheimpflug camera. To solve this problem, this paper used two reference targets which can build translational and rotational models for the Scheimpflug camera to reduce the impact of camera motion.

Figure 5 depicts the camera motion that consists of translation and rotation. The translation in the *z* direction of the camera can be ignored compared to the measured distance. Because the size of the camera in the *z* direction is larger than its size in the *x* and *y* directions, when the camera is firmly fixed, its rotation around the *z*-axis is significantly small, which can also be ignored. The translation in the *x* and *y* directions causes additional displacement errors, *ct_x_* and *ct_y_,* in the image coordinate. The rotation around the *x*-axis and *y*-axis causes additional displacement errors, *cr_x_* and *cr_y_*, in the image coordinate. Here, the *x*, *y* and *z* directions correspond to the horizontal, vertical and depth directions of the Scheimpflug camera, as shown in Figure 4.

Assume that there is a measuring unit (two reference targets and a measuring target located between them) to be measured in the image plane, as shown in Figure 6. The centers of the two reference targets are *A* and *B*, respectively, and the center of the measuring target is *P*, which are obtained by averaging the image coordinates of the two cross-shaped corners on the targets. The scale conversion factors (mm/pixel) of the plane where each target is located are $s_{x}^{A}$, $s_{y}^{A}$, $s_{x}^{P}$, $s_{y}^{P}$, $s_{x}^{B}$ and $s_{y}^{B}$, and the width and height of the image are *W* and *H*, respectively. Therefore, the physical displacements of target *P*, *A* and *B* without camera motion compensation can be expressed by the following formulas:(3)$$\left\{\begin{array}{c} M_{x}^{P-Uncorrected}=M_{x}^{P-Corrected}+ct_{x}^{P}\times s_{x}^{P}+cr_{x}^{P}\times s_{x}^{P}\\ M_{y}^{P-Uncorrected}=M_{y}^{P-Corrected}+ct_{y}^{P}\times s_{y}^{P}+cr_{y}^{P}\times s_{y}^{P} \end{array}\right.,$$
(4)$$\left\{\begin{array}{c} M_{x}^{A}=ct_{x}^{A}\times s_{x}^{A}+cr_{x}^{A}\times s_{x}^{A}\\ M_{x}^{B}=ct_{x}^{B}\times s_{x}^{B}+cr_{x}^{B}\times s_{x}^{B} \end{array}\right.,$$
(5)$$\left\{\begin{array}{c} M_{y}^{A}=ct_{y}^{A}\times s_{y}^{A}+cr_{y}^{A}\times s_{y}^{A}\\ M_{y}^{B}=ct_{y}^{B}\times s_{y}^{B}+cr_{y}^{B}\times s_{y}^{B} \end{array}\right.,$$
where $M_{x}^{P-Corrected}$ and $M_{y}^{P-Corrected}$ are the corrected physical displacements in *x* and *y* directions. The displacements of the two reference targets *A* and *B* are only caused by camera motions.

Errors caused by camera rotation around *x*-axis and *y*-axis

When the camera rotates around the *x*-axis or *y*-axis, it only causes an error in the *y* direction (*cr_y_*) or *x* direction (*cr_x_*), respectively. First, the influence of camera rotation about the *y*-axis on the displacement error was analyzed, as shown in Figure 7. Assuming that the camera rotates clockwise around the focus point *f*, then the rotation angle is *θ*. For demonstration purposes, the rotation of the camera was replaced by rotations of the targets *P*, *A* and *B*. After the rotation, the positions of the targets become *P*′, *A*′ and *B*′. *o* is the center point of the image; $p\left(x_{p},y_{p}\right)$ and $p^{\prime }\left(x_{p^{\prime }},y_{p^{\prime }}\right)$ are the points in the image before and after the rotation of measuring point *P*; $a\left(x_{a},y_{a}\right)$ and $a^{\prime }\left(x_{a^{\prime }},y_{a^{\prime }}\right)$ are the points in the image before and after the rotation of reference point *A*; $b\left(x_{b},y_{b}\right)$ and $b^{\prime }\left(x_{b^{\prime }},y_{b^{\prime }}\right)$ are the points in the image before and after the rotation of reference point *B*. When analyzing the changes in the targets induced by the rotation of the Scheimpflug camera, the tilt angle (*α*) of the sensor should be considered; thus, the changes in the targets should be discussed under two conditions.

(1) When the image sensor tilts right (the definitions of left and right are depicted in Figure 4), the changes in the targets *P*, *A* and *B* in the image coordinate can be derived from Figure 7a:(6)$$cr_{x}^{P}=\left\{\begin{array}{c} \frac{l_{fp}\times \tan\;\theta \times \cos∠ofp}{s\times \cos\left(∠ofp+\alpha \right)},\;\;\left(x_{p}-\frac{W}{2}\right)>0\\ \frac{l_{fp}\times \tan\;\theta \times \cos∠ofp}{s\times \cos\left(∠ofp-\alpha \right)},\;\;\left(x_{p}-\frac{W}{2}\right)\leq 0 \end{array}\right.,\;cr_{x}^{A}=\left\{\begin{array}{c} \frac{l_{fa}\times \tan\;\theta \times \cos∠ofa}{s\times \cos\left(∠ofa+\alpha \right)},\;\;\left(x_{a}-\frac{W}{2}\right)>0\\ \frac{l_{fa}\times \tan\;\theta \times \cos∠ofa}{s\times \cos\left(∠ofa-\alpha \right)},\;\;\left(x_{a}-\frac{W}{2}\right)\leq 0 \end{array}\right.,\;cr_{x}^{B}=\left\{\begin{array}{c} \frac{l_{fb}\times \tan\;\theta \times \cos∠ofb}{s\times \cos\left(∠ofb+\alpha \right)},\;\;\left(x_{b}-\frac{W}{2}\right)>0\\ \frac{l_{fb}\times \tan\;\theta \times \cos∠ofb}{s\times \cos\left(∠ofb-\alpha \right)},\;\;\left(x_{b}-\frac{W}{2}\right)\leq 0 \end{array}\right.,$$
where the rotation angle *θ* is regarded as significantly small because the camera rotation is significantly small in practice. *s* is the pixel size (3.45 µm/pixel in this study), which is equal in the *x* and *y* directions. $l_{fp}=f/\cos\;∠ofp$, $l_{fa}=f/\cos\;∠ofa$, $l_{fb}=f/\cos\;∠ofb$, $∠ofp=\arctan\;\left(\left(\left|x_{p}-W/2\right|\times s\right)/f\right)$, $∠ofa=\arctan\;(\left(\left|x_{a}-W/2\right|\times s\right)/f)$, $∠ofb=\arctan\;(\left(\left|x_{b}-W/2\right|\times s\right)/f)$, *f* is the focal length.

Here, the proportion between the changes in the two reference targets is as follows:(7)$$\frac{cr_{x}^{A}}{cr_{x}^{B}}=\left\{\begin{array}{c} \begin{array}{c} \frac{\cos\;\left(∠ofb+\alpha \right)}{\cos\;\left(∠ofa+\alpha \right)},\;\;\left(x_{a}-\frac{W}{2}\right)>0\;and\;\left(x_{b}-\frac{W}{2}\right)>0\\ \frac{\cos\;\left(∠ofb-\alpha \right)}{\cos\;\left(∠ofa-\alpha \right)},\;\;\left(x_{a}-\frac{W}{2}\right)\leq 0\;and\;\left(x_{b}-\frac{W}{2}\right)\leq 0 \end{array}\\ \begin{array}{c} \frac{\cos\;\left(∠ofb+\alpha \right)}{\cos\;\left(∠ofa-\alpha \right)},\;\;\left(x_{a}-\frac{W}{2}\right)>0\;and\;\left(x_{b}-\frac{W}{2}\right)\leq 0\\ \frac{\cos\;\left(∠ofb-\alpha \right)}{\cos\;\left(∠ofa+\alpha \right)},\;\;\left(x_{a}-\frac{W}{2}\right)\leq 0\;and\;\left(x_{b}-\frac{W}{2}\right)>0 \end{array} \end{array}\right.,$$
and the proportion between the changes in measuring target *P* and reference target *A* is as follows:(8)$$\frac{cr_{x}^{A}}{cr_{x}^{P}}=\left\{\begin{array}{c} \begin{array}{c} \frac{\cos\;\left(∠ofp+\alpha \right)}{\cos\;\left(∠ofa+\alpha \right)},\;\;\left(x_{a}-\frac{W}{2}\right)>0\;and\;\left(x_{p}-\frac{W}{2}\right)>0\\ \frac{\cos\;\left(∠ofp-\alpha \right)}{\cos\;\left(∠ofa-\alpha \right)},\;\;\left(x_{a}-\frac{W}{2}\right)\leq 0\;and\;\left(x_{p}-\frac{W}{2}\right)\leq 0 \end{array}\\ \begin{array}{c} \frac{\cos\;\left(∠ofp+\alpha \right)}{\cos\;\left(∠ofa-\alpha \right)},\;\;\left(x_{a}-\frac{W}{2}\right)>0\;and\;\left(x_{p}-\frac{W}{2}\right)\leq 0\\ \frac{\cos\;\left(∠ofp-\alpha \right)}{\cos\;\left(∠ofa+\alpha \right)},\;\;\left(x_{a}-\frac{W}{2}\right)\leq 0\;and\;\left(x_{p}-\frac{W}{2}\right)>0 \end{array} \end{array}\right.,$$

(2) When the image sensor tilts left, the changes in the targets *P*, *A* and *B* in the image coordinate can be derived from Figure 7b.
(9)$$cr_{x}^{P}=\left\{\begin{array}{c} \frac{l_{fp}\times \tan\;\theta \times \cos∠ofp}{s\times \cos\left(∠ofp-\alpha \right)},\;\;\left(x_{p}-\frac{W}{2}\right)>0\\ \frac{l_{fp}\times \tan\;\theta \times \cos∠ofp}{s\times \cos\left(∠ofp+\alpha \right)},\;\;\left(x_{p}-\frac{W}{2}\right)\leq 0 \end{array}\right.,\;cr_{x}^{A}=\left\{\begin{array}{c} \frac{l_{fa}\times \tan\;\theta \times \cos∠ofa}{s\times \cos\left(∠ofa-\alpha \right)},\;\;\left(x_{a}-\frac{W}{2}\right)>0\\ \frac{l_{fa}\times \tan\;\theta \times \cos∠ofa}{s\times \cos\left(∠ofa+\alpha \right)},\;\;\left(x_{a}-\frac{W}{2}\right)\leq 0 \end{array}\right.,\;cr_{x}^{B}=\left\{\begin{array}{c} \frac{l_{fb}\times \tan\;\theta \times \cos∠ofb}{s\times \cos\left(∠ofb-\alpha \right)},\;\;\left(x_{b}-\frac{W}{2}\right)>0\\ \frac{l_{fb}\times \tan\;\theta \times \cos∠ofb}{s\times \cos\left(∠ofb+\alpha \right)},\;\;\left(x_{b}-\frac{W}{2}\right)\leq 0 \end{array}\right.,$$
in this case, the proportion between the changes of the two reference targets is expressed as:(10)$$\frac{cr_{x}^{A}}{cr_{x}^{B}}=\left\{\begin{array}{c} \begin{array}{c} \frac{\cos\;\left(∠ofb-\alpha \right)}{\cos\;\left(∠ofa-\alpha \right)},\;\;\left(x_{a}-\frac{W}{2}\right)>0\;and\;\left(x_{b}-\frac{W}{2}\right)>0\\ \frac{\cos\;\left(∠ofb+\alpha \right)}{\cos\;\left(∠ofa+\alpha \right)},\;\;\left(x_{a}-\frac{W}{2}\right)\leq 0\;and\;\left(x_{b}-\frac{W}{2}\right)\leq 0 \end{array}\\ \begin{array}{c} \frac{\cos\;\;\left(∠ofb-\alpha \right)}{\cos\;\left(∠ofa+\alpha \right)},\;\;\left(x_{a}-\frac{W}{2}\right)>0\;and\;\left(x_{b}-\frac{W}{2}\right)\leq 0\\ \frac{\cos\;\left(∠ofb+\alpha \right)}{\cos\;\left(∠ofa-\alpha \right)},\;\;\left(x_{a}-\frac{W}{2}\right)\leq 0\;and\;\left(x_{b}-\frac{W}{2}\right)>0 \end{array} \end{array}\right.,$$
and the proportion between the changes in measuring target *P* and reference target *A* is as follows:(11)$$\frac{cr_{x}^{A}}{cr_{x}^{P}}=\left\{\begin{array}{c} \begin{array}{c} \frac{\cos\;\left(∠ofp-\alpha \right)}{\cos\;\left(∠ofa-\alpha \right)},\;\;\left(x_{a}-\frac{W}{2}\right)>0\;and\;\left(x_{p}-\frac{W}{2}\right)>0\\ \frac{\cos\;\left(∠ofp+\alpha \right)}{\cos\;\left(∠ofa+\alpha \right)},\;\;\left(x_{a}-\frac{W}{2}\right)\leq 0\;and\;\left(x_{p}-\frac{W}{2}\right)\leq 0 \end{array}\\ \begin{array}{c} \frac{\cos\;\left(∠ofp-\alpha \right)}{\cos\;\left(∠ofa+\alpha \right)},\;\;\left(x_{a}-\frac{W}{2}\right)>0\;and\;\left(x_{p}-\frac{W}{2}\right)\leq 0\\ \frac{\cos\;\left(∠ofp+\alpha \right)}{\cos\;\left(∠ofa-\alpha \right)},\;\;\left(x_{a}-\frac{W}{2}\right)\leq 0\;and\;\left(x_{p}-\frac{W}{2}\right)>0 \end{array} \end{array}\right.,$$

**Figure 7 sensors-22-04093-f007:**
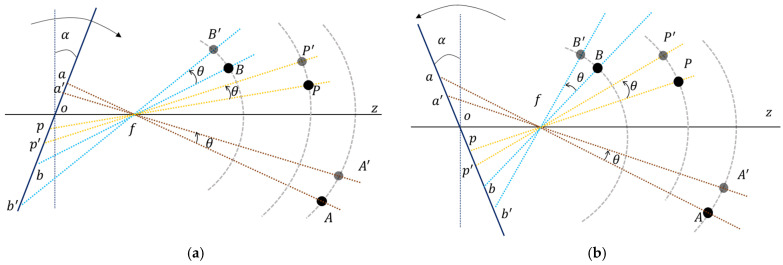
Changes of the targets before and after the camera rotation about the *y*-axis. (**a**) Tilt the sensor (image plane) right. (**b**) Tile the sensor (image plane) left.

Similarly, when the camera rotates around the *x*-axis, the proportion between the changes in the targets in the image coordinate can be derived.

(1) When the image sensor tilts to the right:(12)$$\frac{cr_{y}^{A}}{cr_{y}^{B}}=\left\{\begin{array}{c} \begin{array}{c} \frac{\cos\;\left(∠ofb+\alpha \right)}{\cos\;\left(∠ofa+\alpha \right)},\;\;\left(x_{a}-\frac{W}{2}\right)>0\;and\;\left(x_{b}-\frac{W}{2}\right)>0\\ \frac{\cos\;\;\left(∠ofb-\alpha \right)}{\cos\;\left(∠ofa-\alpha \right)},\;\;\left(x_{a}-\frac{W}{2}\right)\leq 0\;and\;\left(x_{b}-\frac{W}{2}\right)\leq 0 \end{array}\\ \begin{array}{c} \frac{\cos\;\left(∠ofb+\alpha \right)}{\cos\;\left(∠ofa-\alpha \right)},\;\;\left(x_{a}-\frac{W}{2}\right)>0\;and\;\left(x_{b}-\frac{W}{2}\right)\leq 0\\ \frac{\cos\;\left(∠ofb-\alpha \right)}{\cos\;\left(∠ofa+\alpha \right)},\;\;\left(x_{a}-\frac{W}{2}\right)\leq 0\;and\;\left(x_{b}-\frac{W}{2}\right)>0 \end{array} \end{array}\right.,$$
(13)$$\frac{cr_{y}^{A}}{cr_{y}^{P}}=\left\{\begin{array}{c} \begin{array}{c} \frac{\cos\;\left(∠ofp+\alpha \right)}{\cos\;\left(∠ofa+\alpha \right)},\;\;\left(x_{a}-\frac{W}{2}\right)>0\;and\;\left(x_{p}-\frac{W}{2}\right)>0\\ \frac{\cos\;\left(∠ofp-\alpha \right)}{\cos\;\left(∠ofa-\alpha \right)},\;\;\left(x_{a}-\frac{W}{2}\right)\leq 0\;and\;\left(x_{p}-\frac{W}{2}\right)\leq 0 \end{array}\\ \begin{array}{c} \frac{\cos\;\left(∠ofp+\alpha \right)}{\cos\;\left(∠ofa-\alpha \right)},\;\;\left(x_{a}-\frac{W}{2}\right)>0\;and\;\left(x_{p}-\frac{W}{2}\right)\leq 0\\ \frac{\cos\;\left(∠ofp-\alpha \right)}{\cos\;\left(∠ofa+\alpha \right)},\;\;\left(x_{a}-\frac{W}{2}\right)\leq 0\;and\;\left(x_{p}-\frac{W}{2}\right)>0 \end{array} \end{array}\right.,$$
where $∠ofp=\arctan\;((\left|y_{p}-H/2\right|\times s)/f)$, $∠ofa=\arctan\;(\left(\left|y_{a}-H/2\right|\times s\right)/f)$, $∠ofb=\arctan\;(\left(\left|y_{b}-H/2\right|\times s\right)/f)$.

(2) When the image sensor tilts to the left:(14)$$\frac{cr_{y}^{A}}{cr_{y}^{B}}=\left\{\begin{array}{c} \begin{array}{c} \frac{\cos\;\left(∠ofb-\alpha \right)}{\cos\;\left(∠ofa-\alpha \right)},\;\;\left(x_{a}-\frac{W}{2}\right)>0\;and\;\left(x_{b}-\frac{W}{2}\right)>0\\ \frac{\cos\;\left(∠ofb+\alpha \right)}{\cos\;\left(∠ofa+\alpha \right)},\;\;\left(x_{a}-\frac{W}{2}\right)\leq 0\;and\;\left(x_{b}-\frac{W}{2}\right)\leq 0 \end{array}\\ \begin{array}{c} \frac{\cos\;\left(∠ofb-\alpha \right)}{\cos\;\left(∠ofa+\alpha \right)},\;\;\left(x_{a}-\frac{W}{2}\right)>0\;and\;\left(x_{b}-\frac{W}{2}\right)\leq 0\\ \frac{\cos\;\left(∠ofb+\alpha \right)}{\cos\;\left(∠ofa-\alpha \right)},\;\;\left(x_{a}-\frac{W}{2}\right)\leq 0\;and\;\left(x_{b}-\frac{W}{2}\right)>0 \end{array} \end{array}\right.,$$
(15)$$\frac{cr_{y}^{A}}{cr_{y}^{P}}=\left\{\begin{array}{c} \begin{array}{c} \frac{\cos\;\left(∠ofp-\alpha \right)}{\cos\;\left(∠ofa-\alpha \right)},\;\;\left(x_{a}-\frac{W}{2}\right)>0\;and\;\left(x_{p}-\frac{W}{2}\right)>0\\ \frac{\cos\;\left(∠ofp+\alpha \right)}{\cos\;\left(∠ofa+\alpha \right)},\;\;\left(x_{a}-\frac{W}{2}\right)\leq 0\;and\;\left(x_{p}-\frac{W}{2}\right)\leq 0 \end{array}\\ \begin{array}{c} \frac{\cos\;\left(∠ofp-\alpha \right)}{\cos\;\left(∠ofa+\alpha \right)},\;\;\left(x_{a}-\frac{W}{2}\right)>0\;and\;\left(x_{p}-\frac{W}{2}\right)\leq 0\\ \frac{\cos\;\left(∠ofp+\alpha \right)}{\cos\;\left(∠ofa-\alpha \right)},\;\;\left(x_{a}-\frac{W}{2}\right)\leq 0\;and\;\left(x_{p}-\frac{W}{2}\right)>0 \end{array} \end{array}\right..$$

2.Errors caused by camera translation along *x* and *y* directions

When the camera translates in the *x* and *y* directions, it only causes an error in the *x* direction (*ct_x_*) or *y* direction (*ct_y_*). First, the influence of the camera translation in the *y* direction on the displacement error was analyzed, as shown in Figure 8. Assuming that the translation amount of the camera along the *y* direction is Δ*_t_*, the translation of the camera is also replaced by the translation of the targets *P*, *A* and *B*. After the translation, the positions of the targets become *P*′, *A*′ and *B*′, and their changes in the image coordinate can be derived from Figure 8:(16)$$\left\{\begin{array}{c} ct_{y}^{P}=op-op^{\prime }\\ ct_{y}^{A}=oa-oa^{\prime }\\ ct_{y}^{B}=ob-ob^{\prime }\; \end{array}\right..$$

Several experiments show that the error induced by camera translation is significantly smaller than that induced by rotation, and thus the influence of the tilt angle (*α*) on camera translation can be ignored. That is, assuming that the image plane is perpendicular to the lens axis, the proportion between the changes in the targets can be approximately expressed as (15):(17)$$\frac{ct_{y}^{A}}{ct_{y}^{B}}=\frac{d_{B}}{d_{A}},\;\frac{ct_{y}^{A}}{ct_{y}^{P}}=\frac{d_{P}}{d_{A}},$$
where *d_P_*, *d_A_* and *d_B_* represent the physical distance between the target and the camera. However, it is difficult to directly measure the distance between the target and the camera; therefore, the scale conversion factor of the plane where the target is located is used to replace the target-camera distance. It is known that there is a linear positive correlation between them; thus, the following formula can be obtained:(18)$$\frac{ct_{y}^{A}}{ct_{y}^{B}}=\frac{d_{B}}{d_{A}}=\frac{s_{y}^{B}}{s_{y}^{A}},$$
(19)$$\frac{ct_{y}^{A}}{ct_{y}^{P}}=\frac{d_{P}}{d_{A}}=\frac{s_{y}^{P}}{s_{y}^{A}}.$$

Similarly, when the camera translates along the *x* direction, the changes in the two reference targets in the image coordinate have the following proportions:(20)$$\frac{ct_{x}^{A}}{ct_{x}^{B}}=\frac{s_{x}^{B}}{s_{x}^{A}},$$
(21)$$\frac{ct_{x}^{A}}{ct_{x}^{P}}=\frac{s_{x}^{P}}{s_{x}^{A}}.$$

**Figure 8 sensors-22-04093-f008:**
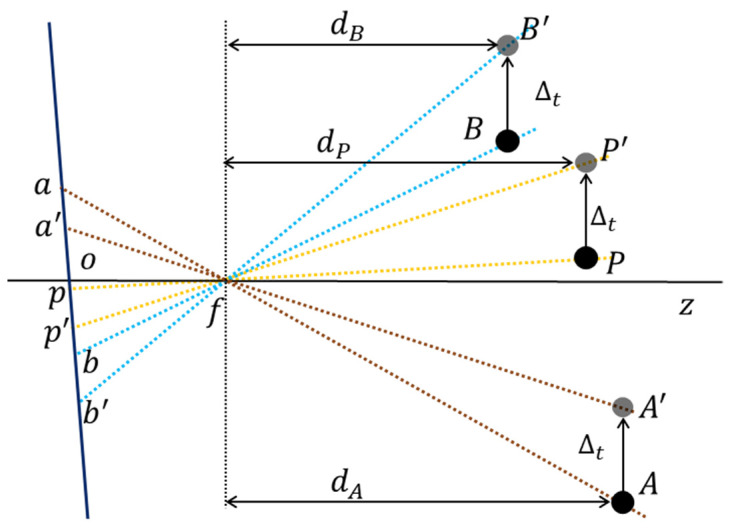
Changes of the targets before and after the camera translation in the *y* direction.

3.Displacement calculation with camera motion compensation

Firstly, Equations (4) and (5) can be simplified by using Equations (18) and (20):(22)$$\left\{\begin{array}{c} M_{x}^{B}-M_{x}^{A}=cr_{x}^{B}\times s_{x}^{B}-cr_{x}^{A}\times s_{x}^{A}\\ M_{y}^{B}-M_{y}^{A}=cr_{y}^{B}\times s_{y}^{B}-cr_{y}^{A}\times s_{y}^{A}\; \end{array},\right.$$
according to different tilt directions of the image sensor, different equations are used to calculate the error components of the targets *A* and *B*: (1) When the image sensor tilts right, $cr_{x}^{A}$, $cr_{y}^{A}$, $cr_{x}^{B}$ and $cr_{y}^{B}$ can be calculated by substituting Equations (7) and (12) into Equation (22); or (2) when the image sensor tilts left, $cr_{x}^{A}$, $cr_{y}^{A}$, $cr_{x}^{B}$ and $cr_{y}^{B}$ can be calculated by substituting Equations (10) and (14) into Equation (22). Secondly, $ct_{x}^{A}$, $ct_{x}^{B}$, $ct_{y}^{A}$ and $ct_{y}^{B}$ can be computed easily through Equations (4) and (5). Then, the error components of the target *P* $\left(ct_{x}^{P},cr_{x}^{P},ct_{y}^{P},cr_{y}^{P}\right)$ can be calculated using Equations (8), (13), (19) and (21) or Equations (11), (15), (19) and (21) according to the tilt direction of the image sensor. Finally, Equation (3) can be used to compute the corrected physical displacement of target *P*. 

4.Measurement stage
(1)After installation of the Scheimpflug camera, one must read the tilt angle (*α*) of the image sensor on the Scheimpflug adapter (the resolution of the adapter is 0.1°) and determine the tilt direction. When the measurement starts, the tilt angle and direction of the image sensor remain unchanged.(2)The Scheimpflug camera measures (*x_p_*, *y_p_*)*_i_*, (*x_a_*, *y_a_*)*_i_* and (*x_b_*, *y_b_*)*_i_* for each target in the *i*-th image.(3)The uncorrected physical displacements ${\left(M_{x}^{P-Uncorrected}\;,M_{y}^{P-Uncorrected}\right)}_{i}$, ${\left(M_{x}^{A}\;,M_{y}^{A}\right)}_{i}$ and ${\left(M_{x}^{B}\;,M_{y}^{B}\right)}_{i}$ in the *i*-th image are calculated with respect to the reference image.(4)According to the tilt direction of the image sensor, the error components ${\left(cr_{x}^{A}\;,cr_{y}^{A},ct_{x}^{A}\;,ct_{y}^{A}\right)}_{i}$ and ${\left(cr_{x}^{B}\;,cr_{y}^{B},ct_{x}^{B}\;,ct_{y}^{B}\right)}_{i}$ of the reference target *A* and *B* in the *i*-th image are calculated by using Equations (4), (5), (7), (12) and (22), or Equations (4), (5), (10), (14) and (22).(5)According to the tilt direction of the image sensor, the error components ${\left(ct_{x}^{P}\;,cr_{x}^{P},ct_{y}^{P}\;,cr_{y}^{P}\right)}_{i}$ of the target *P* in the *i*-th image are calculated by using Equations (8), (13), (19) and (21) or Equations (11), (15), (19) and (21).(6)The corrected physical displacement of the target *P* in the *i*-th image is calculated by using Equation (3). Note that this approach assumes that the out-of-plane motion of the target can be neglected. The displacement calculation process of other measuring targets is the same as that of target *P*.

## 3. Experiment Validation

### 3.1. Validation through a Slide Table Test

The main purpose of this test was to verify the effectiveness of the proposed motion compensation method of the Scheimpflug camera. As shown in Figure 9, the Scheimpflug camera installed on a si*x*-axis slide table observed four fixed targets aligned along the depth direction, in which the si*x*-axis slide table was used to simulate the camera motions. The distance between the camera and the nearest target (No. 105) was 6.1 m. The distance between two adjacent targets was 0.6 m. These four targets constituted a measurement unit mentioned in Figure 3, where targets 105 and 285 were regarded as reference targets to compensate for the camera motions. When the tilt direction was to the right and tilt angle *α* was approximately 5.1°, the camera could clearly capture the four targets (Figure 9c). If the image plane was parallel to the lens plane, the camera could capture only one or two targets clearly (Figure 9d).

Four acquisitions were performed in this test, and the sampling rate of the camera was set to 2 frames per second. In the first two acquisitions, the translation motions of the camera were simulated by slowly translating the slide table in the *x* and *y* directions, while in the last two acquisitions, the rotation motions of the camera were simulated by slowly rotating the slide table around its *y*-axis and *x*-axis, and 150 images were collected in each acquisition. The four targets were fixed in this test, whose real displacements can be considered as zeros. Correspondingly, the displacements detected by the camera were namely the displacement measurement errors induced by camera motions.

Since targets 105 and 285 were reference targets whose displacements defaulted to zeros, and the measurements of target 225 were highly consistent with those of target 165, only the displacement of target 165 is plotted in Figure 10. In the first two acquisitions, the translation of the camera was close to 10 mm, while in the last two acquisitions, the rotation of the camera exceeded 1°, and caused a displacement error of more than 25 mm. After the compensation, the maximum errors of the four results did not exceed 0.01 mm. 

Considering that the resolution of the Scheimpflug adapter was only 0.1°, five different tilt angles {4.9°, 5.0°, 5.1°, 5.2°, 5.3°} were used to verify the influence of the adapter resolution on camera motion compensation. Taking the third acquisition as an example, the corrected displacements of target 165 obtained with different tilt angles are shown in Figure 11.

It can be seen from Figure 11 that the corrected displacements obtained with five different tilt angles had little differences, and the maximum difference was only 0.01 mm. Therefore, it can be concluded that the error caused by the insufficient adapter resolution has a negligible influence on camera motion compensation.

### 3.2. Outdoor Test Using Static Targets

Five static targets fixed on the ground were monitored in this test, as shown in Figure 12 and Figure 13. The Scheimpflug camera was installed on a 1.0 m tall tripod to enable the targets to be observed horizontally (Figure 12). The targets were well distributed along the road direction such that all the targets were collectively observed in a narrow FOV without lowering the camera magnification. Similar to the first test, in this test, targets 1 and 5 were used as reference targets for camera motion compensation. In bridge monitoring applications, these two reference targets are usually installed on two adjacent piers; therefore, the distance between targets 1 and 5 is the span length of the bridge, which is one of the main factors affecting the measurement accuracy. Consequently, the distance *L* between the two reference targets was set to 20 m, 40 m and 80 m to cover different span lengths. In addition, the distance between the camera and the target is also one of the key factors affecting the measurement accuracy; thus, the distance *d* between the camera and target 1 was set to 50 m and 80 m, respectively. Therefore, a total of six acquisitions were conducted to comprehensively evaluate the effectiveness of the system and method proposed in this study. A comparison of the imaging results between the Scheimpflug camera and the conventional camera is shown in Figure 13.

The sampling rate of the camera was set to 90 frames per second, and the duration of each acquisition was 100 s. The purpose of installing multiple targets in the test was to illustrate the capability of the multi-point displacement measurement of the proposed system. However, because the displacements of targets 2, 3 and 4 were almost the same, owing to space and clarity, only the displacements of target 3 are shown in Figure 14.

In this test, we placed the camera on a busy road (Figure 12); thus, passing cars caused obvious ground vibrations. Moreover, the maximum air velocity on the test day exceeded 8 m/s. Under the combined action of these two factors, there were many sudden variations in the original displacements of target 3. In addition, the original displacements of target 3 also showed a gradual decreasing trend, because the camera was prone to slow movement due to its self-weight and temperature changes. However, these factors did not affect the effectiveness of the proposed method, and the corrected displacements of the six acquisitions obtained satisfactory accuracy. 

The root mean squared errors (RMSEs) with and without compensation were calculated, and are listed in Table 1. After implementing the motion compensation method, the RMSEs in the *x* and *y* directions did not exceed 0.54 mm, which were reduced by at least 62% and 92%, respectively. It can be observed from Table 1 that the reductions of RMSEs in the *y* direction were overall higher than those in the *x* direction, which is because the self-weight of the camera and ground vibrations were more likely to cause the camera’s movement in the *y* direction. In addition, the increase in the camera–target distance reduced the image resolution and localization accuracy of the targets, resulting in a worse correction to motion-induced errors.

This test verified the remote measurement performance of the proposed system and method under outdoor conditions. When the measurement distance *d* and span length *L* were all 80 m, that is, the farthest measurement distance was 160 m, the total RMSE and maximum error reached about 0.6 mm and 1.0 mm, respectively. Therefore, in order to ensure that the measurement accuracy is within 1.0 mm, the proposed system and method can only be applied to bridges with a span length of 160 m or less.

### 3.3. Bridge Model Experiment

The proposed system was implemented on a truss structure bridge model with a length of approximately 38.8 m to measure its dynamic displacements. The geometric configuration of the bridge model to be inspected is shown in Figure 15a,b. The whole bridge model was fixed on four shake tables (STs), which were provided by SERVOTEST [36] and arranged in a straight line. The ST 1–ST 2 distance was 6.54 m, and both the ST 2–ST 3 distance and ST 3–ST 4 distance were 13.08 m. These four STs had the same technical indices. They were all six-axis shake tables with a table size of 4 × 4 m^2^; the maximum payload of a single ST was 30 t; the maximum displacements were 250 mm in *x* and *y* directions and 160 mm in *z* direction; the maximum speed in *x*, *y* and *z* directions was ±1000 mm/s; the operating frequency range was 0.1–50 Hz. Here, the *x*, *y* and *z* directions corresponded to the horizontal, vertical and depth directions, respectively, as shown in Figure 15a,b. In addition, the four STs had a flexible operation mode; that is, they can be used independently or concatenated into a shake table array.

Five targets were used in this experiment: target 1 was located 0.1 m ahead of the front end, and target 5 was located 5.2 m behind the back end of the bridge model. These two targets were not fixed on the bridge model; thus, they were static during the experiment and can be regarded as references for compensating camera motion. Other targets were attached to the bridge model. The size of each target was 300 mm × 200 mm, and the physical length between the two cross-shaped corners was 100 mm. 

As shown in Figure 15c, the test site was very narrow; therefore, it was impractical to find a suitable installation position for a single conventional camera to clearly observe all targets at high magnification. In contrast, the Scheimpflug camera has better practicability in these narrow sites, as shown in Figure 15c. All targets could be clearly observed by installing the Scheimpflug camera near the bridge model (Figure 15e), in which the distance between the Scheimpflug camera and target 1 was 10.8 m.

This experiment simulated the impact of an earthquake on a bridge structure. Figure 16 shows the measurement results of the proposed Scheimpflug camera-based system. Images were captured at 60 frames per second. In the entire process of the experiment, the four STs were concatenated into a shake table array and vibrated synchronously in *x* and *y* directions, so the displacements of targets 2, 3 and 4 had the same varying tendency. The maximum displacement amplitude of target 3 was slightly larger than that of target 2 and 4, which were 5.14 mm and 1.79 mm in *x* and *y* directions, respectively, mainly because target 3 was farthest from the shake table. Note that there remained slight vibrations with a maximum value of 0.36 mm in the *y* direction at target 5; this is because the oil-fired engine driving the shake tables released a significant amount of heat during operation, and the resulting hot-air turbulence caused image deformation and additional measurement errors. The engine was located under the bridge model between target 1 and target 2, so the displacement results of targets 2, 3, 4 and 5 were all affected.

To further validate the proposed Scheimpflug-camera system, the measured displacement of target 3 was compared to the values measured by the LDV-based method. Two LDVs with an accuracy of 0.05 mm were installed near target 3, as shown in Figure 15d. Figure 17 shows that the two displacements shared similar overall trends. The RMSEs of the differences between the two displacements in *x* and *y* directions were 0.16 mm and 0.11 mm, respectively. Thus, the performance of the proposed system and method in measuring the dynamic displacements was verified. However, due to the influence of hot-air turbulence, the maximum differences between the two displacements in *x* and *y* directions were 0.76 mm and 0.41 mm, respectively, which is still far from the requirement for a high-precision measurement. It is desired for the proposed method to capture images on bridges when little difference exists between the air and ground temperatures.

## 4. Discussion

The effectiveness of the proposed system and method has been proven, but the following practical issues should be considered when applying the system to actual bridge monitoring.

(1)Out-of-plane motion of target(2)The proposed motion compensation method does not consider the out-of-plane motion of the target; that is, the displacement of the bridge along the road direction is ignored. However, in practical applications, the out-of-plane motion of the target is inevitable, which causes additional calculation errors of scale conversion factors when high-magnification-ratio images are captured through a super-telephoto lens. This decreases the measurement accuracy of our method. Therefore, the proposed method needs to be further optimized.(3)Placement restrictions in camera installation(4)The camera must be installed close to the bridge. In Figure 15c, the shortest distance between the camera and the bridge model is approximately 1.0 m, and only such a short distance can ensure that all targets can be collected in a narrow camera view. However, when monitoring actual bridges, there may be insufficient installation space in front of the bridge.(5)Image noise, blur and deformation caused by remote measurement(6)As shown in Figure 14, when the span length of the bridge or measurement distance increases, the measurement accuracy reduces significantly owing to the noise, blur and deformation of the image. Unmanned aerial vehicles (UAVs) can provide an opportunity to capture bridge images more effectively by bringing the camera closer to the bridge; thus, the UAV equipped with the Scheimpflug camera can be used to realize the short-distance measurement, so as to further improve the accuracy of the Scheimpflug camera-based technique in bridge monitoring. However, the distance (span length of the bridge) between the two piers for fixing reference targets will still restrict the effectiveness of camera motion compensation, which makes the proposed method difficult to be applied to long-span bridges, such as suspension bridges or cable-stayed bridges.

## 5. Conclusions

In this study, we proposed a low-cost system based on a single Scheimpflug camera to measure displacements at many artificial targets attached to a bridge, such that all targets are clearly observed in a single-camera view without reducing the lens magnification. The existing camera-ego-motion compensation methods using reference targets do not consider the case that the image plane is not perpendicular to the lens axis. To solve this problem, this paper built translational and rotational models for the Scheimpflug camera to reduce the error induced by the Scheimpflug-camera motion, which only requires the simple processing of two-dimensional images.

The proposed method was verified through three experiments. In the first experiment, a si*x*-axis slide table was used to simulate camera motions. The maximum error induced by the slide table exceeded 25 mm, and then it was suppressed to 0.01 mm using the proposed method. Regarding outdoor conditions, the performance of the method was verified through different measurement distances and span lengths. The results showed that when the span length of the bridge is no more than 160 m, the measurement accuracy of the proposed system will be better than 1.0 mm. The span length (the distance between two adjacent piers) of the bridge and measurement distance are the two main factors affecting the applicability of the proposed method. Finally, a bridge model experiment was conducted and the performance of the proposed system in measuring the dynamic displacements of bridges was demonstrated. Next, we plan to carry out UAV-related research to prevent the influence of remote measurement by bringing the camera closer to the bridge.

## Figures and Tables

**Figure 1 sensors-22-04093-f001:**
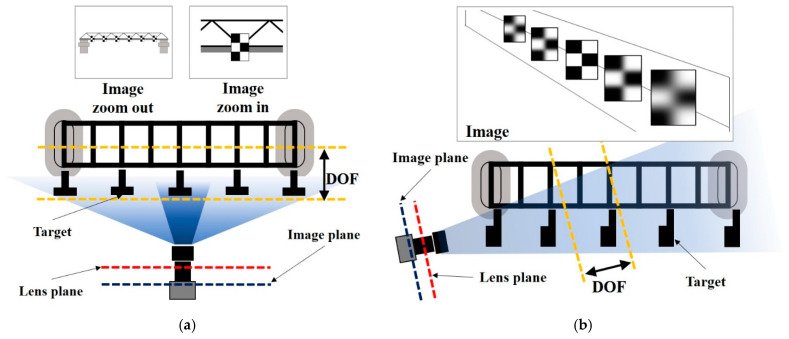
Limitations in vision-based bridge displacement measurement using conventional camera. (**a**) Side-view measurement. (**b**) Front-view measurement.

**Figure 2 sensors-22-04093-f002:**
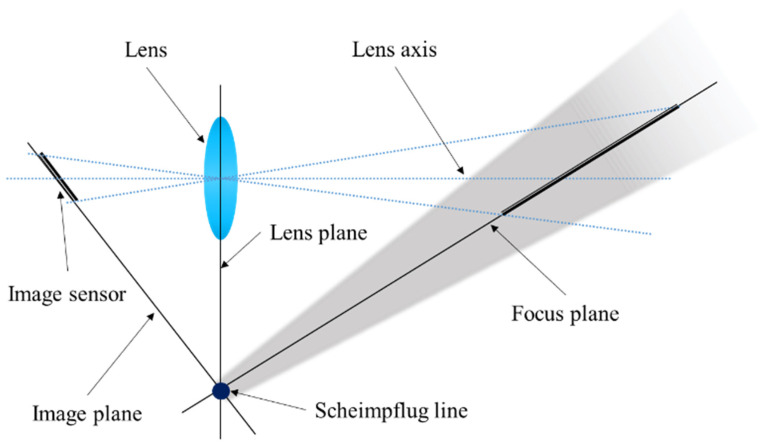
Scheimpflug principle in a 2D view, where gray shade represents DOF.

**Figure 3 sensors-22-04093-f003:**
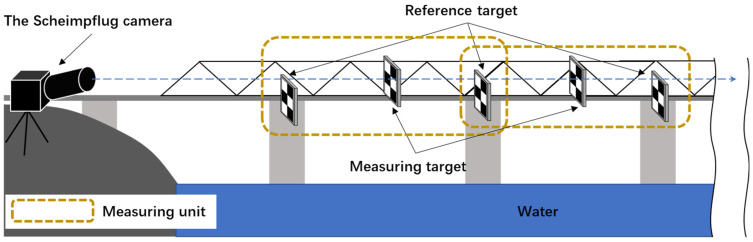
Scheimpflug camera-based system.

**Figure 4 sensors-22-04093-f004:**
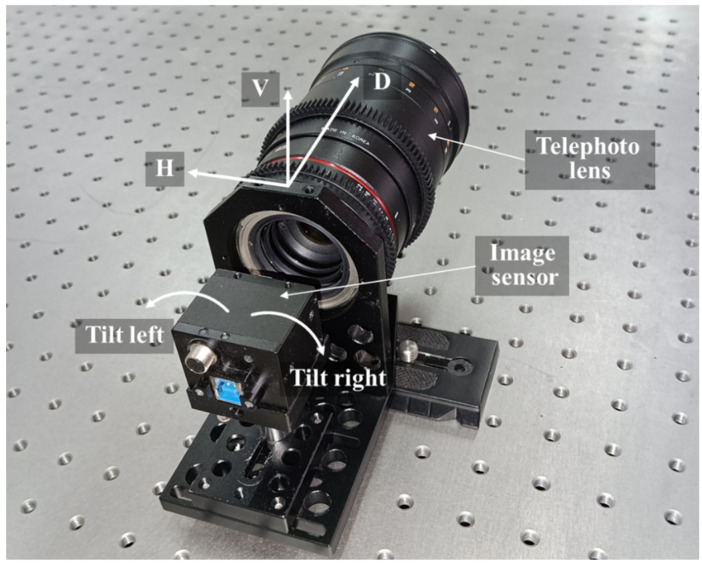
Scheimpflug camera.

**Figure 5 sensors-22-04093-f005:**
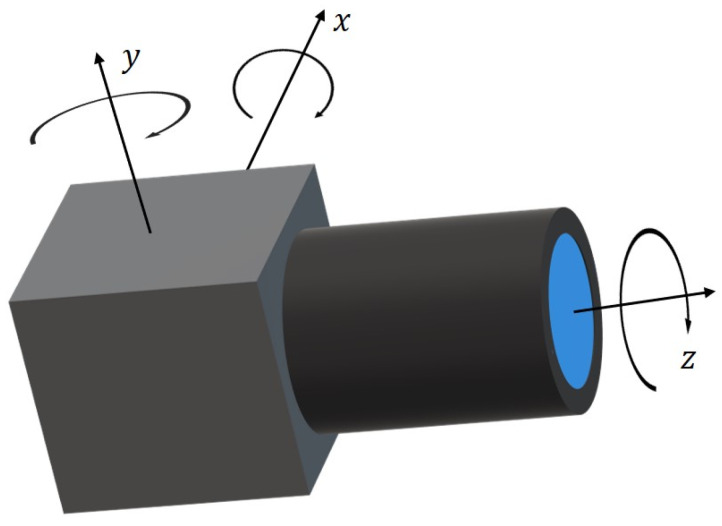
Camera motions during measurement.

**Figure 6 sensors-22-04093-f006:**
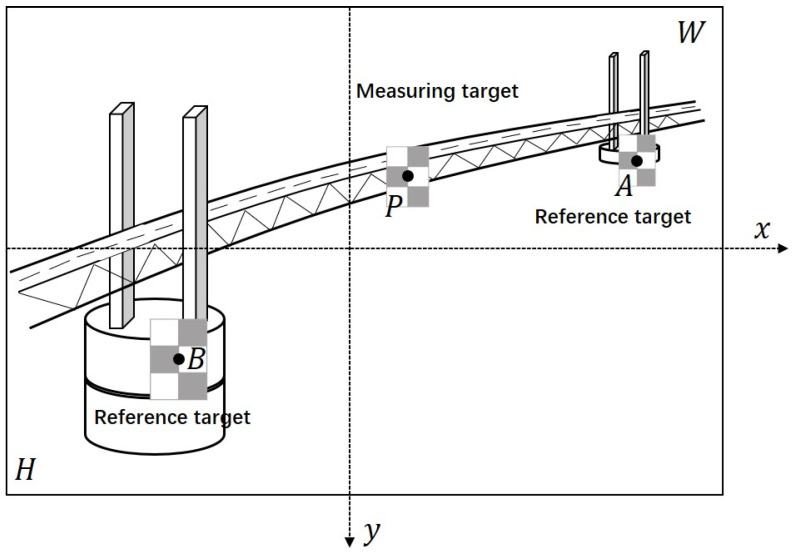
Measuring unit in the image plane.

**Figure 9 sensors-22-04093-f009:**
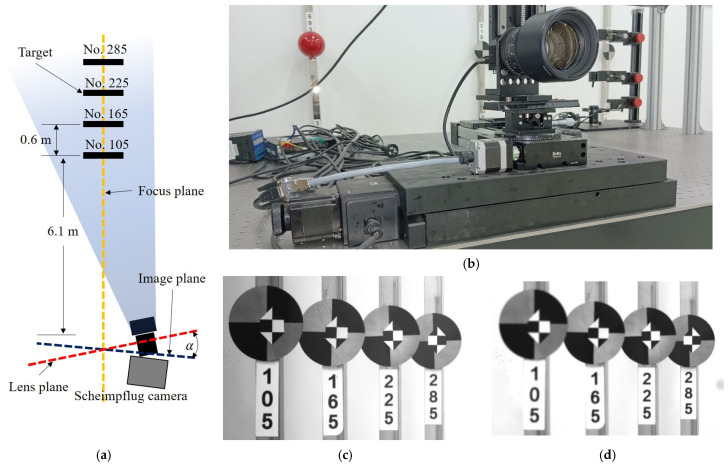
Experiment setup and target selection. (**a**) Experiment setup, where blue shade represents FOV. (**b**) Scheimpflug camera. (**c**) The imaging result of the Scheimpflug camera. (**d**) The imaging result of the conventional camera.

**Figure 10 sensors-22-04093-f010:**
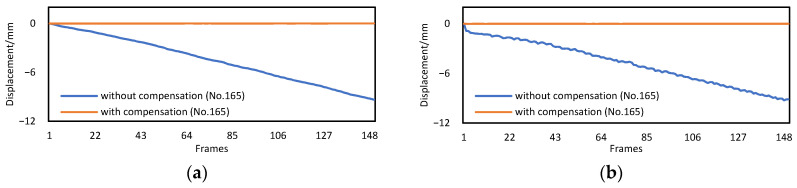
The measurement results obtained by the Scheimpflug camera without/with motion compensation. (**a**) Displacement results of camera translation in the *x* direction. (**b**) Displacement results of camera translation in the *y* direction. (**c**) Displacement results of camera rotation around the *x*-axis. (**d**) Displacement results of camera rotation around the *y*-axis.

**Figure 11 sensors-22-04093-f011:**
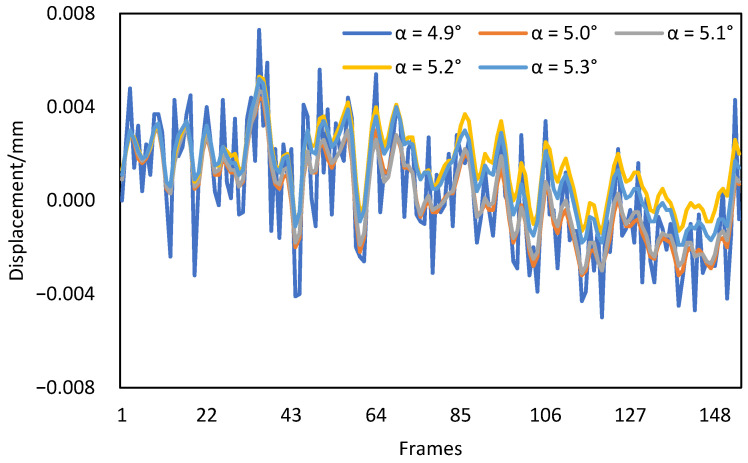
The corrected displacements of target 165 obtained with different tilt angles.

**Figure 12 sensors-22-04093-f012:**
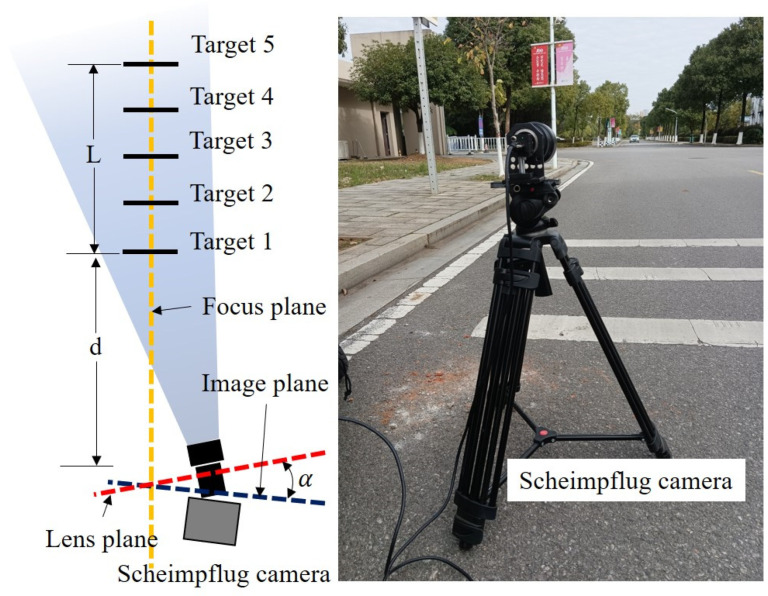
Experiment setup of the outdoor test.

**Figure 13 sensors-22-04093-f013:**
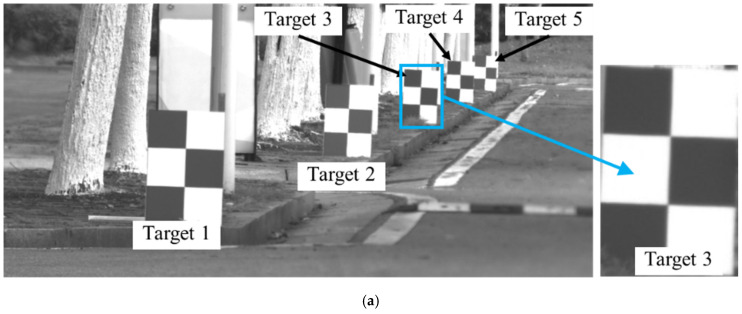
A comparison of the imaging results between the Scheimpflug camera and the conventional camera. (**a**) The imaging result of the Scheimpflug camera. (**b**) The imaging result of the conventional camera.

**Figure 14 sensors-22-04093-f014:**
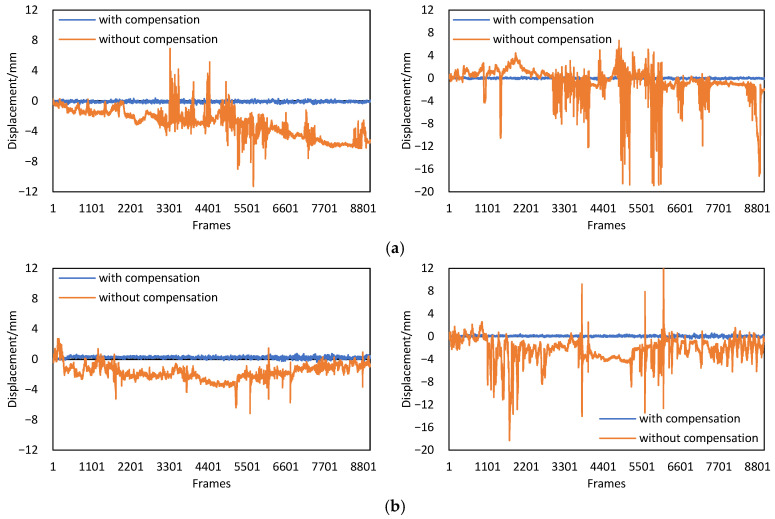
Displacement results of target 3 in the *x* (**left column**) direction and *y* (**right column**) direction. (**a**) *d* = 50 m, *L* = 20 m. (**b**) *d* = 50 m, *L* = 40 m. (**c**) *d* = 50 m, *L* = 80 m. (**d**) *d* = 80 m, *L* = 20 m. (**e**) *d* = 80 m, *L* = 40 m. (**f**) *d* = 80 m, *L* = 80 m.

**Figure 15 sensors-22-04093-f015:**
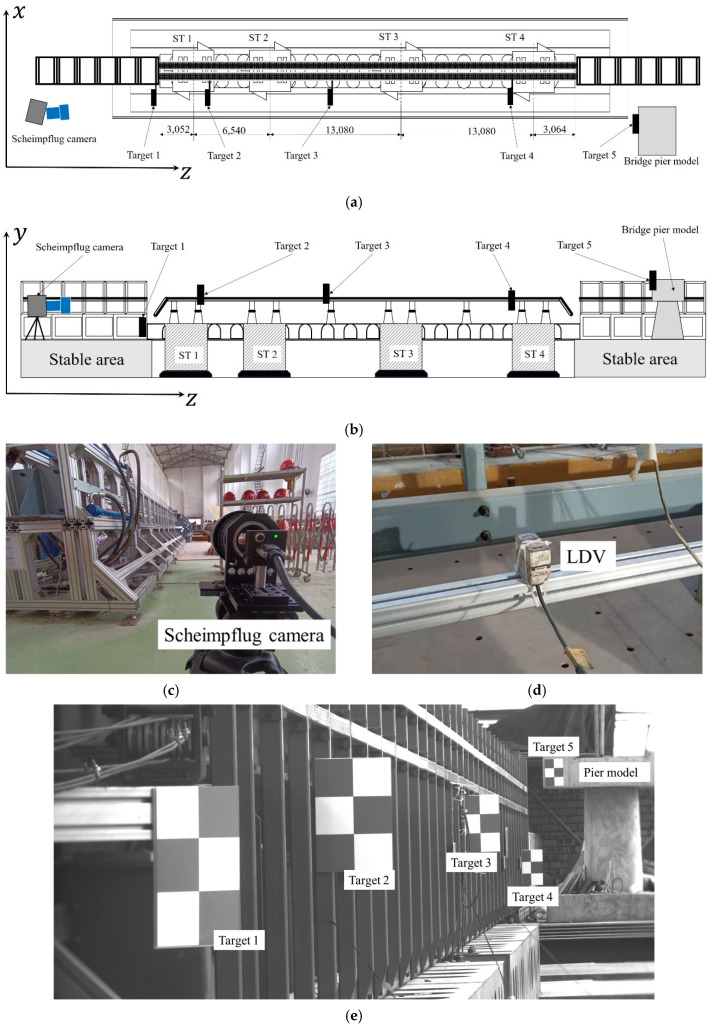
Experiment setup of the bridge model experiment. (**a**) Geometric configuration in the bridge model experiment (**top-view**). (**b**) Geometric configuration in the bridge model experiment (**side-view**). (**c**) The Scheimpflug camera. (**d**) The LDV installed at the bottom of the model. (**e**) An example of images captured by the Scheimpflug camera.

**Figure 16 sensors-22-04093-f016:**
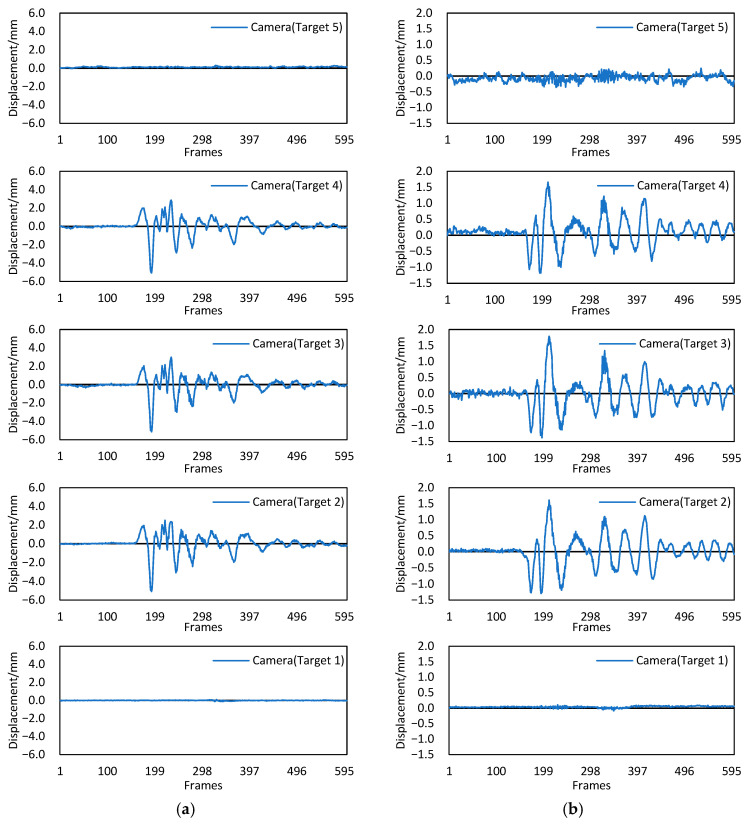
Dynamic displacements from 5 targets. (**a**) Displacements along the *x* direction. (**b**) Displacements along the *y* direction.

**Figure 17 sensors-22-04093-f017:**
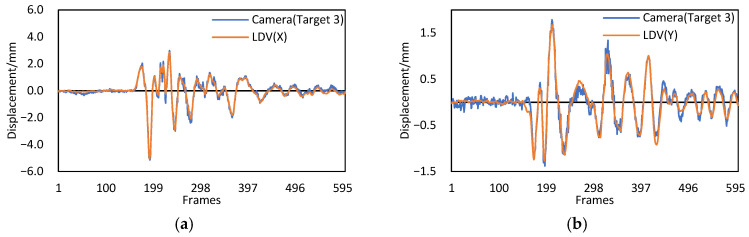
Comparison of the displacements with the Scheimpflug camera and LDV. (**a**) Displacements in the *x* direction. (**b**) Displacements in the *y* direction.

**Table 1 sensors-22-04093-t001:** RMSEs with/without camera motion compensation.

Acquisitions	With Compensation (mm)		Without Compensation (mm)		Reduction in RMSE (%)	
	*x*	*y*	*x*	*y*	*x*	*y*
*d* = 50 m, *L* = 20 m	0.14	0.09	3.44	2.63	96	97
*d* = 50 m, *L* = 40 m	0.27	0.14	1.97	3.12	86	96
*d* = 50 m, *L* = 80 m	0.21	0.20	2.01	7.13	90	97
*d* = 80 m, *L* = 20 m	0.21	0.19	1.78	4.25	88	95
*d* = 80 m, *L* = 40 m	0.50	0.29	1.31	3.43	62	92
*d* = 80 m, *L* = 80 m	0.33	0.54	2.31	8.27	86	93

## Data Availability

Not applicable.

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
