# Peer review of "Scheimpflug Camera-Based Technique for Multi-Point Displacement Monitoring of Bridges"

_sensors, 2022, doi:10.3390/s22114093_

Round 1

Reviewer 1 Report

Dear authors,

you have written very interesting article according to my opinion, and to my field of interest. English language is fine and doesn’t need any improvements. You didn’t cite all the literature. I didn’t find any major flaws in the article, i.e., in the methodology, presentation and results.

The good points of the article are:

  1. very interesting topic regarding bridge monitoring using contactless techniques in multi-points on the bridge spans, i.e., bridge – congrats!
  2. very detailly derived model for motion compensation of the camera – congrats!

According to my opinion, the weak points of your article are:

  1. it is not clear is this approach applicable for large-scale bridges.
  2. design of the article
  3. all references are not cited in the article

The improvement should be done in the following:

Ad. 1) It is not clear is this approach applicable for large-scale bridges. Further, in the whole paper you always state large-scale bridges. You should specify exactly what is large scale bridge (up to 100 m, 100-300 m, 300-500 m, 500-1000 m, or larger). This is confusing because when you read the article one could expect that this camera and approach is applicable for large bridges. And in the end, you state, which is correct according to the results, that when the distance increases the measurement accuracy significantly reduces… This should be cleared, i.e., I suggest that in introduction and abstract you clearly state what is the aim and task of this study regarding its application and on which bridges.

Ad 2.) The design of the article according to my opinion should be rearranged as follows:

  • Section 2 and 3 should be merged to one section 2 and named Materials and methods
  • In this new section 2, old section 2 will become subsection 2.1 and old section 3 will become subsection 2.2
  • all old sections that follows should be renumbered

Ad. 3)

The references 12 and 27 are not cited in the article. I couldn’t find them. Please have a look at it and correct if necessary.

Suggestion by “lines”:

- line 30 – please add geodetic instruments – total stations and image assisted total stations. Please have a look at three studies: https://doi.org/10.3390/app11093893, https://doi.org/10.3390/s21237952, https://doi.org/10.3390/geomatics2010001

- line 33 - GNSS is limited by the accessibility to the structures – this is not totally true. If GNSS is limited with this, then every other instrument or sensor is limited also with this fact. GNSS is limited with its accuracy in real time (3-5 cm in vertical direction) and with the fact that it needs clear horizon.

- line 34 – reference is needed for this.

- line 49 – total station is not a device – it is a geodetic instrument

- line 102 – please correct fro+nt to front

- line 259 - The resolution of the adapter is 0.1° How big is the influence of the adapter resolution on camera movement compensation?

- line 289 – 2 fps, line 323 – 90 fps, line 373 – 60 fps – why did you use different fps in tests?

- line 362 – mm2 – mm?

Best regards.

Reviewer 2 Report

This paper presents empirical research results to increase the measurement accuracy by the motion compensation method with Scheimpflug camera. It has been verified through 3 real test results. It meets the purpose of the sensors journal, and the limitations of this method such as out-of-plane motion of target and placement restrictions are also reasonably presented.

1. It is necessary to explain results of other numbers except for No. 165 target shown in Figure 10.

2. It seems that an additional explanation is needed for the cause of the noise in Figure 13.

3. The units of numbers in Table 1 are presented and more explanations seem to be necessary.

4. In Figure 15, the results by existing image-based measurement should be presented together to confirm the accuracy of this study.
